# Protective Effects of Dietary MUFAs Mediating Metabolites against Hypertension Risk in the Korean Genome and Epidemiology Study

**DOI:** 10.3390/nu11081928

**Published:** 2019-08-16

**Authors:** Hansongyi Lee, Han Byul Jang, Min-Gyu Yoo, Kyung-Sook Chung, Hye-Ja Lee

**Affiliations:** 1Center for Biomedical Sciences, Korea National Institute of Health, Cheongju 28159, Korea; 2Department of Pharmaceutical Biochemistry, Kyung Hee University, Seoul 02447, Korea

**Keywords:** monounsaturated fatty acid /MUFAs, metabolites, hypertension

## Abstract

Background and Aims: Metabolites related to dietary factors can be used to identify biological markers to prevent metabolic disease. However, most studies have been conducted in the United States and Europe, and those in the Asian region are limited. We investigated the effects of dietary monounsaturated fatty acids (MUFAs) and metabolites on new-onset hypertension in the Korean Genome and Epidemiology Study. Method and Results: A total of 1529 subjects without hypertension were divided into tertiles of dietary MUFAs intake. After a 4-year follow-up, 135 serum metabolites were measured using the AbsoluteIDQ p180 kit. During the 4-year follow-up period, 193 new-onset hypertension incidences were observed. The highest MUFAs intake group was inversely associated with the risk of hypertension compared with the lowest MUFAs intake group (odds ratio (OR) = 0.49, (95% confidence interval (CI) = 0.29–0.82)). Of the 135 metabolites, eight were significantly associated with MUFAs intake. Phosphatidylcholine-diacyl (PC aa) C 38:1 and hydroxysphingomyelin (SM OH) C 16:1 were associated with a decrease in hypertension risk (PC aa C 38:1, OR = 0.60 (95% CI = 0.37–0.96); SM OH C 16:1, OR = 0.42 (95% CI = 0.20–0.90)). The highest MUFAs intake group had a significantly decreased risk of hypertension, even considering PC aa C 38:1 and SM (OH) C 16:1 as a mediator. Conclusion: We confirmed that dietary MUFAs intake, and PC aa C 38:1 and SM (OH) C 16:1 had protective effects against hypertension. Furthermore, high MUFAs intake combined with PC aa C 38:1 and SM (OH) C 16:1 has the most significant effect on reducing the risk hypertension.

## 1. Introduction

Hypertension is the most prevalent and modifiable risk factor for cardiovascular disease. According to the most recent Korean National Health and Nutrition Examination Survey, 31.2% of adults have hypertension and only 43.4% of these have controlled hypertension [1]. A decrease in blood pressure of 3 mm Hg reduces coronary death by 5% and stroke death by 8% [2]. Hypotensive therapies combined with modified diet or mineral supplements such as zinc have been recently recommended [3,4,5] to moderate the oxidative status, glucose status or insulin resistance associated with hypertension.

Investigating novel biomarkers of disease prevention or management is a preventive strategy. Metabolites represent the intermediate and end products of cellular processes and are important for signaling, structural membranes, and catalytic activity. Several cohort studies have reported that some metabolites are associated with the incidence of hypertension. According to the European Prospective Investigation into Cancer and Nutrition-Potsdam study, four metabolites, serine, glycine, phosphatidylcholine (PC)-acyl-alkyl (ae), and PC-diacyl (aa) are associated with the incidence of hypertension [6]. The Atherosclerosis Risk in Communities study determined that metabolites (4-hydroxyhippurate) or metabolite patterns (sex steroid pattern) were associated with incident hypertension during a 10-year follow-up [7]. The Twins UK study identified a novel pathway (ω-oxidation) for blood pressure regulation involving hexadecanedioate [8]. 

Several studies have also identified metabolites associated with food intake and related metabolic diseases. Dietary protein, or a healthy diet (alternative healthy eating index) have been shown to regulate type 2 diabetes [9], renal disease [10], and cardiovascular disease [11] through amino acids and lipid metabolites. The PREDIMED trial reported that a Mediterranean diet may mitigate the adverse associations between higher concentrations of plasma branched chain amino acids [12], glutamate [13], acylcarnitines [14], ceramides [15], and incident cardiovascular disease. Several dietary factors (dietary protein and vegetable intake) or dietary interventions (e.g., a carbohydrate, protein, or MUFAs-rich diet, the DASH diet) have been used to regulate blood pressure through changes in metabolites such as alanine, hippurate, proline betaine, carnitine, and 4-ethylphenylsufate in urine or blood [16,17,18]. 

Monounsaturated fatty acids (MUFAs) have either neutral or hypotensive effects on blood pressure when compared to carbohydrate-rich [19,20,21], saturated fatty acids [22,23], or polyunsaturated fatty acid diets [24,25]. A few mechanisms have been associated with MUFAs, particularly that mechanism whereby oleic acid regulates membrane lipid structure to control G-protein-mediated signaling and reduce blood pressure [26]. 

To our knowledge, few studies have investigated the association between metabolites and chronic disease in South Korea [27,28]. Therefore, the purpose of this study was to examine the effects of dietary MUFAs and metabolites against new-onset hypertension in the Korean Genome and Epidemiology Study (KoGES).

## 2. Material and Methods 

### 2.1. Study Population and Database

The KoGES is an ongoing series of prospective cohort studies begun in 2003–2004 by the Korean National Institute of Health and Centers for Disease Control to identify gene–environment factors and their interactions in common chronic disease, such as diabetes, hypertension, metabolic syndrome, obesity and CVD. The overall data were drawn from the Ansan-Ansung population cohort study, a community-based study of the KoGES. Subjects’ metabolite data (*n* = 2580) were included at the second follow-up (2005–2006). Among these, subjects with missing MUFAs consumption data at baseline (*n* = 146), hypertension (*n* = 846), cardiovascular disease (*n* = 58), or cancer (*n* = 1) were excluded. Ultimately, 1529 subjects were included in the final analysis. 

The study protocol was approved by the Institutional Ethical Review Board of the Korea Centers for Disease Control and Prevention (no. 2017-02-06-2C-A). 

### 2.2. MUFAs Intake Measurements

The dietary intake assessment including MUFAs has been described in detail previously [29]. Briefly, dietary assessments were collected using a validated semi-quantitative food frequency questionnaire to check the consumption frequency and serving size of 103 food items. Average daily nutrient intake was calculated as consumption frequency per serving × portion size for each food item. Dietary MUFAs were estimated using the food composition table published by the Rural Development Administration of Korea [30]. 

### 2.3. Definition of Hypertension 

Professionally trained personnel performed blood pressure measurements using a standardized protocol. The subject sat for 10 min and their blood pressure was measured three times at intervals of 5 min each using a mercury sphygmomanometer at baseline and biannually. The mean blood pressure value was used to define hypertension. Hypertension was defined based on the Joint National Committee criteria: systolic blood pressure ≥140 mmHg or diastolic blood pressure ≥90 mmHg, or taking antihypertensive medication during follow-up.

### 2.4. Metabolite Quantification

Data on serum metabolites were drawn from the KoGES sample analyses, which were performed according to the manufacturer’s instructions and described elsewhere [27]. Briefly, serum samples for metabolite analyses were collected from 2580 subjects after the 4-year follow-up. Liquid chromatography and flow injection analysis mass spectrometry were performed to quantify the metabolites using the AbsoluteIDQ p180 kit (Biocrates Life Sciences AG, Innsbruck, Austria). Ultimately, 135 metabolites met the study criteria, including 13 acylcarnitines, 21 amino acids, 10 biogenic amines, 34 PCs aa C, 36C PCs ae C, 8 lysoPCs, 12 sphingomyelins (SMs), and 1 hexose. 

### 2.5. Blood Parameters and Covariates

Fasting plasma glucose, total cholesterol, triglycerides, and high-density lipoprotein (HDL) cholesterol were measured using a Hitachi 747 chemistry analyzer (Hitachi Ltd., Tokyo, Japan) following the manufacturer’s recommendations. Demographic characteristics (age, sex, household income, and education) medical history (hypertension, diabetes, cardiovascular disease, and cancer), and lifestyle factors (smoking, drinking status, and physical activity) were obtained using questionnaires administered during interviews. Physical activity levels were calculated from the self-reported duration and intensity of physical activity considering metabolic equivalents. Body mass index (BMI) was calculated as weight (kg) divided by height squared (m^2^). 

### 2.6. Statistical Analysis 

Prior to the analyses, dietary MUFAs intake was adjusted for total energy intake according to the residual method, and the metabolite values were highly skewed, thus these values were normalized by logarithmic transformation. Comparisons of the baseline variables across the MUFAs tertiles were analyzed using the chi-squared test for categorical variables. Continuous variables were analyzed using one-way analysis of variance and significant differences in the means were detected by Duncan’s post-hoc analysis. Values are expressed as the mean ± standard deviation for continuous variables and numbers (%) for categorical variables. To select metabolites, multiple linear regression models between MUFAs intake and metabolites were obtained after adjusting for confounding variables (age, sex, energy intake, BMI, metabolic equivalents, smoking status, drinking status, household income, education, and diabetes status). Additionally, the false discovery rate (FDR)-corrected *p*-value defined by the Benjamini–Hochberg method was used to control for the effect of multiple testing. We obtained the odds ratios (ORs) for hypertension risk according to MUFAs or metabolite intake using a multivariate logistic regression model adjusted for confounding variables. All statistical analyses were performed using Statistical Analysis System software (SAS version 9.4, SAS Institute, Cary, NC, USA). For all tests, *p*-value < 0.05 was considered to be a statistically significant difference.

## 3. Results

Table 1 shows the baseline characteristics of the subjects according to MUFAs intake. The subjects in the highest MUFAs intake group were significantly younger and a greater proportion were male than those in the other groups (*p* < 0.05). BMI did not differ according to the MUFAs intake group. Body fat percent, waist, and waist to hip ratio were highest in the lowest MUFAs intake groups compared to other groups (*p* < 0.05). Systolic blood pressure decreased significantly with increasing MUFAs intake. The mean value of the lowest MUFAs intake group was 115.1 ± 11.9 mmHg, which was higher than other groups (middle MUFAs intake group and the highest MUFAs intake group, 112.1 ± 11.6 mmHg and 111.8 ± 11.8 mmHg, respectively) (*p* < 0.0001). Fasting blood glucose concentrations, total cholesterol, and HDL cholesterol levels differed significantly according to MUFAs intake (*p* < 0.05). The renin value did not differ according to MUFAs intake. Subjects in the highest MUFAs group had a higher education level and household income, and were more likely to be current smokers or drinkers compared to subjects in the other groups (*p* < 0.001).

During the 4-year follow-up period, 193 new-onset hypertension cases were observed. After adjusting for age, sex, energy intake, BMI, metabolic equivalents, smoking status, drinking status, household income, education, and diabetes status, logistic regression revealed that the highest MUFAs intake group was inversely associated with the incidence of new-onset hypertension compared to the lowest intake group (OR = 0.42, 95% CI = 0.29–0.82) (Table 2). The values for systolic and diastolic blood pressure decreased according to MUFAs intake (*p* < 0.05) (Table 3).

Next, we investigated whether MUFAs intake was associated with metabolite concentration (Table 4). Of the 135 metabolites, 8 were positively associated with MUFAs intake (FDR-adjusted *p* < 0.05). Higher MUFAs intake was associated with two higher PCs aa C (PC aa C 38:0 and PC aa C 38:1) (β = 0.0063–0.0075), five PCs ae C (PC ae C 36:0, PC ae C 38:5, PC ae C 40:4, PC ae C 40:6 and PC ae C 42:5) (β = 0.0040–0.0065), and one hydroxyl-SM (SM OH C 16:1) (β = 0.0055). These eight metabolites were further investigated as potential mediators.

Multivariate logistic regression analyses were performed to determine whether these metabolites were associated with hypertension (Table 5). Among the eight metabolites, PC aa C 38:1 and SM OH C 16:1 were associated with a decreased hypertension risk (PC aa C38:1, OR = 0.60 (95% CI = 0.37–0.96); SM OH C 16:1, OR = 0.42 (95% CI = 0.20–0.90)).

Also, we evaluated the association between hypertension incidence and metabolites combined with dietary MUFAs intake (Table 6). The highest MUFAs intake group had a significantly decreased hypertension risk in association with low or high PC aa C 38:1 concentration (low, OR = 0.45 (95% CI = 0.23–0.95); high, OR = 0.39 (95% CI = 0.20–0.78)) as well as SM (OH) C 16:1 concentration (low, OR = 0.40 (95% CI = 0.20–0.83); high, OR = 0.44 (95% CI = 0.23–0.86)) compared to the reference group. 

## 4. Discussion

We first identified eight out of 135 metabolites (two PCs aa C (38:0, 38:1), five PCs ae C (36:0, 38:5, 40:4, 40:6, and 42:5) and one SM OH C 16:1) that were associated with MUFAs intake. Of these, PC aa C 38:1 and SM (OH) C 16:1 were associated with reduced hypertension risk. Although MUFAs intake was related to two metabolites, PC aa C 38:1 and SM (OH) C 16:1, which reduced hypertension risk, high MUFAs intake was strongly associated with reduced hypertension risk. 

Accumulating evidence has shown that dietary MUFAs are associated with cardiometabolic risk factors (such as blood pressure, hypertension, or CVD incidence). However, most of these studies have been conducted in the United States and Europe, and limited information is available for the Asian region. Notably, in the present study, the high MUFAs intake reduced the incidence of hypertension risk by about 51% in South Korea. In line with our results, several cohort or cross-sectional studies [31,32,33] and meta-analyses [34,35] have shown that dietary MUFAs have beneficial effects on cardiometabolic risk factors.

Several factors including caffeine, gut microbiomes, salt, etc. could be the link between metabolites and the hypertension. The four metabolites of caffeine (namely 3-methlyuric acids, 7-methlyuric acids, 3-methylxanthines and 7-methylxanthines) significantly reduced hypertension risk [36]. Short chain fatty acids produced by gut microbiota modulated blood pressure via olfactory receptor 78 and G-coupled receptor 41 [37]. Hypertension-related sodium intake showed an association with changes in metabolites such as 4-ethylphenylsufate [18]. Also, the previous study indicated that trace mineral supplementation had a favorable effect on mineral status and glucose status [4].

In the present study, we first ascertained that dietary MUFAs could be the link between metabolites and the new onset of hypertension. Previously, dietary polyunsaturated fatty acids have been thought to explain the possible relationship between metabolites and hypertension through regulation of the renin-angiotensin II system [38]. The renin values in this study were not affected by the degree of MUFAs consumption, suggesting mechanisms other than PUFAs might be involved [39]. A possible mechanism for dietary MUFAs lowering hypertension risk may be the modified composition of the phospholipid membrane or vascular reactivity, and the net effects of this may either raise or lower blood pressure [40,41]. Thus, we identified that phospholipids, such as PC and SM (as plasma membrane components) increased along with the increase in MUFAs. PC or SM and their metabolites play key roles in regulating the pathogenesis of metabolic abnormalities. For instance, five PC metabolites (PC aa C 36:5, 36:6, 38:5, 38:6, and 40:6) are associated with obesity and type 2 diabetes based on FTO genotype in the Korean population [27]. PC inhibits up-regulation of inflammatory cytokines, such as tumor necrosis factor-α, interleukin-6, and macrophages [42]. On the other hand, the SM metabolites (SM C 16:0, 16:1, and 24:0) are linked to coronary heart disease risk [43] or a higher risk of myocardial infarction [44].

Regarding hypertension, higher PC aa C 38:3 and 38:4 have been associated with a lower 10-year hypertension prevalence [6], and PC aa C 34:4 has been significantly associated with increased diastolic blood pressure but not with incident hypertension [45]. We preliminarily confirmed that PCs aa C 38:3 and 38:4 were associated with the prevalence of hypertension (PC aa C 38:3, OR = 3.33 (95% CI = 1.62–6.83), FDR = 0.006; PC aa C 38:4, OR = 2.66 (95% CI = 1.29–5.49), FDR = 0.034) and PC aa C 34:4 significantly increased diastolic blood pressure (β = 0.00498, SE = 0.00107, *p* ≤ 0.0001). However, for the metabolite selection, by considering dietary MUFAs and our results, we ultimately selected PC aa C 38:1 and SM (OH) C 16:1 out of the eight metabolites. Interestingly, PC aa C 38:1 had a protective effect against hypertension risk (~40% reduction), along with high MUFAs intake (~51% reduction). However, combined with MUFAs intake, low and high PC aa C 38:1 concentration produced similar reductions in hypertension (by about 55% and 61%, respectively). Similarly, SM (OH) C 16:1 reduced hypertension risk (by about 58%), and combined with MUFAs intake, low or high SM (OH) C 16:1 concentrations had similar effects against hypertension (a reduction of about 60% and 56%, respectively). Taken together, these data suggest that both high MUFAs intake and the two metabolites were protective factors against hypertension risk; however, high MUFAs intake was more strongly associated with a reduction in hypertension risk. Our findings provide further insight into changes in dietary MUFAs and PC and SM metabolites and the pathway to hypertension, although further research is required.

The current study suggests that high MUFAs intake significantly decreased hypertension risk, and may be a strong protective factor against hypertension. Until now, MUFAs consumption has been recommended as <20% of total energy intake for a healthy adult or one with diabetes [46,47]. However, in Asia, there is no specific optimal recommendation for MUFAs, and there are no data related to incident hypertension mediated with metabolites. Our findings were informative and provided an observational study among general and clinical population; however, there is a need for intervention studies on metabolites. This study had several limitations that should be considered. First, dietary information relied on estimates of habitual consumption over the past year in food frequency questionnaires. Food frequency questionnaires show good reliability and relative validity for assessing dietary intake [48]. Second, plasma metabolite profiles varied between men and women, as do several key pathways involved in hypertension. However, our primary aim was to investigate the association between metabolites and hypertension. Therefore, in our study, we adjusted all models for age, sex, and their interaction. The strength of the present study is that it is the first study to evaluate a large set of metabolites as potential mediators of the association between MUFAs intake and hypertension risk.

## 5. Conclusions 

In conclusion, we first identified eight out of 135 metabolites that were associated with MUFAs intake. Of these, PC aa C 38:1 and SM (OH) C 16:1 reduced hypertension risk. Although these two metabolites and high MUFAs intake reduced hypertension risk, high MUFAs intake was strongly protective against hypertension risk. These findings may help identify biomarkers underlying the association between MUFAs and hypertension. However, replicating our findings in additional studies in diverse populations is needed. 

## Figures and Tables

**Table 1 nutrients-11-01928-t001:** Baseline characteristics according to energy-adjusted * dietary MUFAs intake among KoGES study participants (*n* = 1529).

	Tertiles of Energy-Adjusted MUFAs Intake			*p* ^†^
	T1 (7.9 g/day)	T2 (12.2 g/day)	T3 (18.5 g/day)	
Age (years)	55.3 ± 9.1 ^a^	50.4 ± 8.1 ^b^	48.4 ± 7.4 ^c^	<0.0001
Sex				<0.0001
Male	173 (11.3)	264 (17.3)	302 (19.8)	
Female	336 (22.0)	246 (16.1)	208 (13.6)	
Body mass index (kg/m^2^)	24.1 ± 3.4	24.2 ± 3.1	24.2 ± 3.0	0.839
Body fat (%)	27.6 ± 7.2 ^a^	25.7 ± 6.9 ^b^	25.0 ± 6.5 ^b^	<0.0001
Waist (cm)	81.8 ± 9.1	80.9 ± 8.2	80.9 ± 8.4	0.1617
Waist to hip ratio	0.90 ± 0.1 ^a^	0.89 ± 0.0 ^b^	0.89 ± 0.0 ^b^	<0.0001
Systolic blood pressure (mmHg)	115.1 ± 11.9 ^a^	112.1 ± 11.6 ^b^	111.8 ± 11.8 ^b^	<0.0001
Diastolic blood pressure (mmHg)	76.2 ± 7.8	75.0 ± 8.2	75.3 ± 8.7	0.075
Fasting glucose (mg/dL)	85.1 ± 16.2 ^b^	85.8 ± 15.5 ^b^	88.2 ± 17.1 ^a^	0.006
Triglycerides (mg/dL)	155.1 ± 95.7	151.5 ± 98.6	152.3 ± 101.1	0.831
Total cholesterol (mg/dL)	188.5 ± 36.1 ^b^	193.4 ± 35.6 ^ab^	194.1 ± 33.1 ^a^	0.022
HDL cholesterol (mg/dL)	44.3 ± 9.6 ^b^	45.7 ± 10.2 ^ab^	46.1 ± 10.8 ^a^	0.017
LDL cholesterol (mg/dL)	117.5 ± 34.5	121.4 ± 33.0	121.2 ± 32.7	0.111
Renin (ng/mL/h)	2.89 ± 3.1	2.94 ± 2.8	2.80 ± 2.3	0.7166
Metabolic equivalent (MET/month)	11618.1 ± 6770.2 ^a^	9931.0 ± 5889.2 ^b^	9284.5 ± 5821.9 ^b^	<0.0001

KoGES, Korean Genome and Epidemiology Study; HDL, high-density lipoprotein; LDL, low-density lipoprotein; MUFAs, monounsaturated fatty acids. ***** Adjusted for total energy intake according to the residual method and Willet and Stampfer. ^†^ Statistical differences were determined using one-way analysis of variance for continuous variables and the chi-squared test for categorical variables (*p* < 0.05). ^a–c^ Significant differences by Duncan’s post-hoc analysis.

**Table 2 nutrients-11-01928-t002:** Association (OR and 95% CI) for new-onset hypertension according to MUFAs intake by tertiles.

	Tertiles of Energy-Adjusted MUFAs Intake		
	T1	T2	T3
Model 1	Ref	0.96 (0.67–1.37)	0.52 (0.35–0.79)
Model 2	Ref	0.98 (0.65–1.49)	0.49 (0.29–0.82)

MUFAs, monounsaturated fatty acids; OR, odds ratio. ORs were obtained with a multivariate logistic regression model. Model 1 was adjusted for age, sex, and energy intake. Model 2 was adjusted for Model 1 plus BMI, metabolic equivalents, smoking status, drinking status, household income, education, and diabetes status.

**Table 3 nutrients-11-01928-t003:** Blood pressure after 4-year follow-up according to energy-adjusted * dietary MUFAs intake.

	Tertiles of Energy-Adjusted MUFAs Intake			*p* ^†^
	T1 (7.9 g/day)	T2 (12.2 g/day)	T3 (18.5 g/day)	
Systolic blood pressure (mmHg)	114.0 ± 15.5 ^a^	112.1 ± 13.5 ^b^	110.8 ± 13.1 ^b^	0.0015
Diastolic blood pressure (mmHg)	75.9 ± 9.4 ^a^	76.3 ± 9.8 ^ab^	74.7 ± 8.9 ^b^	0.0198

***** Adjusted for total energy intake according to the residual method and Willet and Stampfer. ^†^ Statistical differences were determined using one-way analysis of variance for continuous variables and the chi-squared test for categorical variables (*p* < 0.05). ^a–c^ Significant differences by Duncan’s post-hoc analysis.

**Table 4 nutrients-11-01928-t004:** Relationship between metabolites and MUFAs intake.

Metabolites	Beta *	SE	*p*-Value	^‡^*q*-Value
PC aa C 38:0	0.0063	0.0018	0.0003	0.017
PC aa C 38:1	0.0075	0.0024	0.0020	0.041
PC ae C 36:0	0.0065	0.0021	0.0017	0.041
PC ae C 38:5	0.0053	0.0015	0.0005	0.017
PC ae C 40:4	0.0050	0.0015	0.0005	0.017
PC ae C 40:6	0.0051	0.0017	0.0021	0.041
PC ae C 42:5	0.0040	0.0013	0.0026	0.044
SM OH C 16:1	0.0055	0.0015	0.0003	0.017

PC aa, phosphatidylcholine diacyl; PC ae, phosphatidylcholine acyl-alkyl; SM OH, hydroxysphingomyelin. ***** Multiple linear regression models were obtained after adjusting for age, sex, energy, BMI, metabolic equivalent, smoking status, drinking status, household income, education, and diabetes status. ^‡^ FDR corrected *p*-value defined by the Benjamini–Hochberg method (*q* < 0.05).

**Table 5 nutrients-11-01928-t005:** Association between hypertension risk (OR and 95% CI) and metabolites.

Metabolites	OR ^†^ (95% CI)	*p*-Value
PC aa C 38:0	0.97 (0.51–1.83)	0.914
PC aa C 38:1	0.60 (0.37–0.96)	0.032
PC ae C 36:0	0.97 (0.56–1.68)	0.921
PC ae C 38:5	1.24 (0.59–2.60)	0.570
PC ae C 40:4	0.68 (0.31–1.48)	0.327
PC ae C 40:6	0.77 (0.38–1.53)	0.452
PC ae C 42:5	0.70 (0.30–1.66)	0.417
SM OH C 16:1	0.42 (0.20–0.90)	0.026

PC aa, phosphatidylcholine diacyl; PC ae, phosphatidylcholine acyl-alkyl; SM OH, hydroxysphingomyelin. ^†^ Odds ratios (ORs) were obtained with a multivariate logistic regression model adjusted for age, sex, energy, BMI, metabolic equivalent, smoking status, drinking status, household income, education, and diabetes status.

**Table 6 nutrients-11-01928-t006:** Association (OR and 95% CI) between hypertension risk and metabolites combined with dietary MUFAs intake.

	**PC aa C 38:1**	
	**Low**	**High**
Tertiles of energy-adjusted MUFAs intake		
TI	244/54 Ref	181/29 0.731 (0.407–1.314) ^†^
T2	207/44 1.087 (0.644–1.836)	232/27 0.616 (0.333–1.139)
T3	191/17 0.446 (0.228–0.951)	276/22 0.392 (0.198–0.776)
	**SM OH C 16:1**	
	**Low**	**High**
Tertiles of energy-adjusted MUFAs intake		
TI	235/58 Ref	191/25 0.669 (0.392–1.249)
T2	208/41 0.928 (0.541–1.593)	232/30 0.773 (0.434–1.380)
T3	203/19 0.402 (0.195–0.830)	267/20 0.443 (0.228–0.863)

MUFAs, monounsaturated fatty acids; PC aa, phosphatidylcholine diacyl; SM OH, hydroxysphingomyelin. ^†^ Odds ratios (ORs) were obtained with a multivariate logistic regression model adjusted for age, sex, energy, BMI, metabolic equivalent, smoking status, drinking status, household income, education, and diabetes status.

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
