# Peer review of "Protective Effects of Dietary MUFAs Mediating Metabolites against Hypertension Risk in the Korean Genome and Epidemiology Study"

_nutrients, 2019, doi:10.3390/nu11081928_

Round 1

Reviewer 1 Report

The authors have investigated the effects of dietary MUFAs and metabolities against new-onset hyprtension in KoGES. They found that some metabolities associated with MUFAs intake have the effects to reduce the risk of hypertension.

Comments

The design of the study is not clear and confused, the results are not presented well and  for the reviewer is not easy to follow it. 

The main points of criticism are:

What is the KoGES? Are the subjectysrecruted for this cohort normotensive, hypertensive or are subjecys with other cardiovascular disease?
The values of blood pressure of the cohort for the study should be reported.  Furthermore, the values of hormones such as renin, angiotensin and aldosterone should be indicated.
They reported that during 4 year of follow up , they observed new-onset hypertesion cases but the method by which they determined was not mentioned and the values of blood pressure for these subjects were not reported, as wellas the hormones.
Which is the link between the metabolites and the onset of hypertension?

Author Response

We are pleased to submit our revised manuscript (Manuscript number: nutrients -562907) titled "Protective effects of dietary MUFAs mediating metabolites against hypertension risk in the Korean genome and epidemiology study" for publication in Nutrients. We revised our manuscript according to reviewers' comments or suggestions. We deeply appreciate the editor's consideration in giving us a chance of revision. We revised the manuscript in red. What is the KoGES? Are the subjectysrecruted for this cohort normotensive, hypertensive or are subjecys with other cardiovascular disease? - We really apologize for the confusion and thank you very much for your review. - We modified Study population and database in Material and Method section. We inserted the purpose of KoGES. The subjects were normotensive or hypertensive or they have CVD. So, we excluded the subjects with hypertension or CVD at baseline for the present study. Also, we reported the values of blood pressure according to reviewer's recommendation. Moreover, we inserted the result of data analysis for the the values of renin in Table1 and explained it. Unfortunately, angiotensin and aldosterone values are not available. The values of blood pressure of the cohort for the study should be reported. Furthermore, the values of hormones such as renin, angiotensin and aldosterone should be indicated. They reported that during 4 year of follow up , they observed new-onset hypertesion cases but the method by which they determined was not mentioned and the values of blood pressure for these subjects were not reported, as wellas the hormones. - We sorry again for the confusion. - We modified the definition of new-onset hypertension in method section as your recommendation. Also, we inserted the values of blood pressure followed-up during 4 years in Table 3. The follow-up values of renin, angiotensin and aldosterone are not available. Which is the link between the metabolites and the onset of hypertension? -- We discussed in red the link between the metabolites and the onset of hypertension in Discussion section as you recommended.

Reviewer 2 Report

I was honored to review the manuscript entitled "Protective Effects of Dietary MUFAs mediating Metabolites against Hypertension Risk in the Korean Genome and Epidemiology Study " submitted to Nutrients.

The aim of this work was to investigate the association between MUFAs metabolites and hypertension.

Taking into account the multiple studies ongoing in the field this type of study is needed.  I have only few small remarks that authors should adress properly.

I recommend  minor revision.

Points that need correction:

- please provide the list of abbreviations.

- please provide the number of ethical approval

 -  did you perform body composition analisis?

- Introduction and Discussion section needs improvement- please cite: doi: 10.3390/nu10091284. ; doi: 10.26402/jpp.2018.2.13. ; doi: 10.1016/j.jtemb.2018.02.016.

 -  in discussion please provide the study strong points and limitations

Author Response

I was honored to review the manuscript entitled "Protective Effects of Dietary MUFAs mediating Metabolites against Hypertension Risk in the Korean Genome and Epidemiology Study " submitted to Nutrients. The aim of this work was to investigate the association between MUFAs metabolites and hypertension. Taking into account the multiple studies ongoing in the field this type of study is needed. I have only few small remarks that authors should adress properly. -We are pleased to submit our revised manuscript (Manuscript number: nutrients -562907) titled "Protective effects of dietary MUFAs mediating metabolites against hypertension risk in the Korean genome and epidemiology study" for publication in Nutrients. We revised our manuscript according to reviewers' comments or suggestions. We deeply appreciate the editor's consideration in giving us a chance of revision. We revised the manuscript in red. I recommend minor revision. Points that need correction: - We thank you very much for your review. Please provide the list of abbreviations. - We provided the list of abbreviations in red on page 8. Please provide the number of ethical approval. - We had already provided the number of ethical approval(no.2017-02-06-2C-A). Did you perform body composition analisis? - We performed body composition analysis and inserted the results in table 1. Introduction and Discussion section needs improvement- please cite: doi: 10.3390/nu10091284. ; doi: 10.26402/jpp.2018.2.13. ; doi: 10.1016/j.jtemb.2018.02.016. - We improved Introduction and Discussion section with recommended references. In discussion please provide the study strong points and limitations. - We previously represented two limitations of our study and a strong point but it did not being exposed come out. The expression of a strong point was slightly revised.

Round 2

Reviewer 1 Report

The paper is improved anf the authors have  responded to all the questions.